# Preparation and Characterization of Thermal-Insulating Microporous Breathable Al/LLDPE/CaCO_3_ Composite Films

**DOI:** 10.3390/ma16124230

**Published:** 2023-06-07

**Authors:** Jungeon Lee, Sabina Yeasmin, Jae Hoon Jung, Tae Young Kim, Tae Yeong Kwon, Da Yeong Kwon, Jeong Hyun Yeum

**Affiliations:** 1Department of Biofibers and Biomaterials Science, Kyungpook National University, Daegu 41566, Republic of Korea; dlwjddjs2@gmail.com (J.L.); yeasminsabina44@yahoo.com (S.Y.); xodud10301@naver.com (T.Y.K.); 2Department of Carbon Hybrid Fiber Science, Kyungpook National University, Daegu 41566, Republic of Korea; wjdwognsz@naver.com; 3Hans Intech Co., Ltd., Daegu 41243, Republic of Korea; kty7289@naver.com (T.Y.K.); dayoung962001@naver.com (D.Y.K.)

**Keywords:** composites, extrusion molding, aluminum, morphology, breathability, thermal insulation, thermal properties, mechanical properties

## Abstract

Breathable films were prepared based on linear low-density polyethylene (LLDPE), calcium carbonate (CaCO_3_), and aluminum (Al; 0, 2, 4, and 8 wt.%) using extrusion molding at a pilot scale. These films must generally be able to transmit moist vapor through pores (breathability) while maintaining a barrier to liquids; this was accomplished using properly formulated composites containing spherical CaCO_3_ fillers. The presence of LLDPE and CaCO_3_ was confirmed by X-ray diffraction characterization. Fourier-transform infrared spectroscopy results revealed the formation of Al/LLDPE/CaCO_3_ composite films. The melting and crystallization behaviors of the Al/LLDPE/CaCO_3_ composite films were investigated using differential scanning calorimetry. Thermogravimetric analysis results show that the prepared composites exhibited high thermal stability up to 350 °C. Moreover, the results demonstrate that surface morphology and breathability were both influenced by the presence of various Al contents, and their mechanical properties improved with increasing Al concentration. In addition, the results show that the thermal insulation capacity of the films increased after the addition of Al. The composite with 8 wt.% Al showed the highest thermal insulation capacity (34.6%), indicating a new approach to transform composite films into novel advanced materials for use in the fields of wooden house wrapping, electronics, and packaging.

## 1. Introduction

Polyethylene (PE), which accounts for one-third of the global plastic production, is one of the most versatile polymers due to its low density, excellent processability, and low cost. PE is primarily used to prepare films. Low-density PE and linear low-density PE (LLDPE) are the two types of PE used to make commercial films. LLDPE is widely used as packaging film due to its superior mechanical characteristics, including high tensile strength, tear, and impact strength [1]. LLDPE is also used to create breathable films, which are low-cost, highly effective, and environmentally beneficial materials. It is uniformly combined with fillers and used in a variety of products, including diapers, disposable clothing, sanitary napkin back sheets, and wooden house covers. Breathable films have a porous structure with various characteristics, and the solution’s viscosity is thoroughly regulated, which is influenced by molecular weight, concentration, solvent, etc. These films must generally allow moist vapor to pass through pores (breathability) while maintaining a barrier to liquids. Calcium carbonate (CaCO_3_) is inexpensive and commercially available, has a large specific surface area, can affect mechanical properties with strong interaction between polymer matrices [2,3], and provides breathability. Studies on CaCO_3_-filled microporous breathable polyolefin films have been detailed in a number of articles, and the effects of filler size and biaxial stretch ratio on pore shape have been explored [4,5,6,7]. A breathable film composed of LLDPE and CaCO_3_ particles was also recently reported by our lab group [8]. The effect of draw ratio on the film’s crystallinity, morphology, mechanical characteristics, pore-size distribution, and water vapor permeability was explored. It takes a lot of CaCO_3_ to make a breathable film; research suggests that at least 30% of it is needed to construct interconnected series of pores that allow for vapor diffusion.

Widespread applications for polymer–metal composite films include packaging [9,10,11] and, more recently, building and cooling as they are used to cover vacuum insulation panels [12]. Several investigations into the heat-sealing capability of aluminum (Al) have been published. To maximize the heat-sealing parameters of such composite polymeric films, Planes et al. [10] investigated the heat-sealing qualities of composites composed of one PE layer and one to three aluminum-coated polyethylene terephthalate layers. Guo and Fan [13] investigated the heat-sealing characteristics of five distinct polymer–aluminum–polymer composite films, which are common packaging materials for commercial pouch lithium-ion batteries, by identifying and evaluating the primary sealing ability factors.

Technologies for manufacturing polymer films include compression molding, transfer molding, injection molding, extrusion molding, film blowing, and calendering. In the extrusion molding method, which is most commonly used for film production, solid resins in pellets or masterbatches are melted by heating through a hopper to form a thin, uniform melt while transporting them. After passing through a roller, it passes through a T-Die that controls the dimensions of the molded product with a certain shape, and the extruded molded product is cooled and manufactured into a film or sheet. Because polyolefin films are formed from their melts, spherulitic crystal formation predominates. In polyolefin films, large spherulitic crystals impede high extension during stretching, which is necessary for microporosity in polyolefin-based films [14].

In this study, thermal-insulating breathable films were prepared based on LLDPE and CaCO_3_, and Al particles using the extrusion molding method. The influence of Al on the material properties of the Al/LLDPE/CaCO_3_ composite films was investigated. The CaCO_3_ concentration was kept constant (100 wt.% of the polymer weight), and the influence of Al loading was determined. Breathable films are inexpensive polymer-based barriers that control moisture. The novelty and purpose of this study lie in the transformation of a composite film into new advanced thermal-insulating breathable materials applicable in various fields such as wooden house wrapping and electronics. This study also aims to assess the impact of different Al contents on the LLDPE/CaCO_3_ structure, as well as the various material properties of the composite film.

## 2. Materials and Methods

### 2.1. Materials

The LLDPE polymer was purchased from SK geo centric Co., Ltd. (Jong-ro, Jongno-gu, Seoul, Republic of Korea). The material characteristics of this polymer are summarized in Table 1. LLDPE was prepared and used in masterbatch form with CaCO_3_ (Yabashi Lime, Ogaki-shi, Japan) and aluminum (Toyo Aluminium K.K., Osaka, Japan).

### 2.2. Preparation of the Al/LLDPE/CaCO_3_ Extruded Composite Films

The extrusion and calendering processes were combined to create Al/LLDPE/CaCO_3_ composite films. Extrusion was used to create Al/LLDPE/CaCO_3_ extrudates, and calendaring was applied to flatten the extrudates into films to replicate current plastic processing techniques.

LLDPE, CaCO_3_, and Al pellets or masterbatches were introduced to a twin-screw extruder (O-Sung Industrial Co., Ltd., Ansan-si, Republic of Korea) with a screw speed of 200 rpm through a hopper. While being transported, the masterbatches were heated through a hopper to create a thin, uniform melt. The resulting extrudates were calendered using a double-drum roller (O-Sung Industrial Co., Ltd., Ansan-si, Republic of Korea). After passing through the roller, it was passed through a T-Die that controls the dimensions of the molded product with a certain shape, and the extruded molded product was cooled and wound by a roller to be manufactured into a film or sheet. Figure 1 illustrates the preparation of the Al/LLDPE/CaCO_3_ composite films. The films were manufactured using manufacturing equipment (from Hans Intech Co., Ltd., Daegu, Republic of Korea). Details of the process conditions are specified in Table 2.

### 2.3. Instrumental Analysis

The dispersion states and morphologies of the Al/LLDPE/CaCO_3_ composite films were examined using scanning electron microscopy (SEM; SU8220, Hitachi, Tokyo, Japan). Each sample was coated with gold prior to analysis. X-ray diffraction (XRD; D/Max-2500, Rigaku, Tokyo, Japan) and Fourier-transform infrared (FTIR) spectroscopy (Frontier, PerkinElmer, Waltham, MA, USA) were used to confirm the formation of the Al/LLDPE/CaCO_3_ composite films. The thermal properties of the Al/LLDPE/CaCO_3_ composite films were investigated using differential scanning calorimetry (DSC; Q2000, TA Instruments, New Castle, DE, USA). Thermogravimetric analysis (TGA; Q-50, TA Instruments, Seoul, Republic of Korea) was used to measure the thermal properties. According to the specifications of ASTM D638-96 type II [15], the mechanical properties were assessed by applying an Instron 5567 material testing system. All data were estimated based on the average of three sample measurements. To measure the thickness of the film, a thickness gauge (Digital Vernier Caliper, Hando, Seoul, Republic of Korea) was used. The KES-F7 Thermo Labo II (Kato Tech, Kyoto, Japan) test method was used to evaluate the thermal insulation of the composite material. The ASTM E96/E96M-12 [16] standard method was used to measure the water vapor transmission rate (WVTR) [17].

## 3. Results and Discussion

The SEM images of the Al/LLDPE/CaCO3 composite films exhibited microvoid generation (Figure 2). The pore-size distribution histograms and average pore diameter of the films are shown in Figure 3. As evident, the Al/LLDPE/CaCO_3_ composite films possess highly porous structures. As the Al content increases, the pore size of the film slightly increases, and cracks are formed, seemingly due to stretching during the manufacturing process of the composite film. The average particle size of CaCO_3_ is 3.00 µm, and CaCO_3_ is confirmed to be uniformly dispersed on the film surface. As the Al content increases from 0 to 8 wt.%, the average pore diameter increases from 3.85 to 4.13 µm (Figure 3). In particular, all films formed numerous irregular cracks and microsized pores. During stretching in the manufacturing process, the CaCO_3_ and Al particles disperse on the PE polymer matrix, promoting pore formation. Independent of the sample type and loading amount, all films show rougher film surfaces, which causes the microvoid generation and CaCO_3_ appearance on the film surface.

Additionally, increasing the Al loading in the PE films provides a slightly large surface with cracks and pores. Moreover, all the porous films represent networks composed of probably interconnected pores. In general, the interconnectedness of the porous films is a crucial characteristic in increasing vapor transfer rates across wrapping materials. In addition, the pores of different sizes vary slightly depending on the Al concentration. This result also emphasizes the Al aggregation that leads to pores being generally larger in size in the composite film with 8 wt.% Al.

The FTIR spectrum of LLDPE presented characteristic peaks at 2917, 2849, and 1472 cm^−1^ for asymmetric CH_2_ stretching, symmetric CH_2_ stretching, and CH_2_ bending, respectively, and split peaks at 731 and 719 cm^−1^ for CH_2_ rocking [18,19,20]. Calcium carbonate showed characteristic absorption bands, which were typically symmetric stretching and asymmetric stretching at about 1080 and 1400 cm^−1^, respectively, and out-of-plane bending and in-plane bending at about 870 and 700 cm^−1^, respectively [21]. The spectra of the LLDPE/CaCO_3_ composite films (Figure 4) exhibited the characteristic peaks of LLDPE and CaCO_3_.

Generally, the peaks at 3440 and 1048 cm^−1^ for aluminum powder are obtained by the hydroxyl groups (Al–OH) on the aluminum surface because ambient moisture was present on the surface of the aluminum powder [22]. However, in this study, no new peak appears after adding Al (2–8 wt.%) to the LLDPE/CaCO_3_ matrix due to the low content of Al compared with the matrix. The presence of the LLDPE characteristics in the Al/LLDPE/CaCO_3_ composite films is noted with a slight reduction in peak intensity, hence signifying that the chemical structures have been preserved without the formation of appreciable chemical bonds between LLDPE/CaCO_3_ and Al. Compared with that of the LLDPE/CaCO_3_ composites, the peak intensity of the Al/LLDPE/CaCO_3_ composite film at 1472 cm^−1^ is more intense due to the intense stretching vibration of the LLDPE/CaCO_3_ matrix.

The 110 reflections of PE are attributed to the XRD peak at 21.6° (Figure 5). Calcium carbonate containing LLDPE exhibits peaks associated with the presence of the calcite phase, and the peak at 2θ = 29.5° shows a typical crystal peak of CaCO_3_ [23]. Among the peaks, significant intense peaks of aluminum occur at 38° and 44.4° due to the 111 and 200 planes, respectively [24]. Here, the Al peak at 44.4° is observed but not the peak at 38° due to overlapping between LLDPE and CaCO_3_, and the peak intensity varies depending on the Al concentration. Additionally, a slight shift in the peak location of PE is observed after the addition of Al. The main LLDPE/CaCO_3_ characteristic peaks appear to be preserved in all the samples according to the XRD results, showing that the addition of Al particles has little effect on the LLDPE matrix’s overall structure.

Figure 6 shows the TGA thermogram of the LLDPE/CaCO_3_ composite at different Al loadings. The composite show high thermal stability up to 350 °C with slight weight loss, probably due to water removal. Above 350 °C, all the films show similar intense degradation due to LLDPE decomposition [25]. The total weight loss of the thermal degradation process is ~50% up to 362 °C and 70% until 692 °C. The majority of the remaining weight above 500 °C is attributed to CaCO_3_, whose decomposition temperature ranges from 850 to 900 °C [25]. Furthermore, the addition of Al causes an increase in stability, as shown by the decomposition temperature graphs of the composite rising at higher temperatures (700–800 °C).

DSC was used to analyze the crystallization behavior of the Al/LLDPE/CaCO_3_ composite films with varying Al contents (0–8% wt.%). The resulting DSC cooling and heating curves are displayed in Figure 7. Table 3 summarizes the experimental findings in terms of melting temperature (T_m_), crystallization temperature (T_c_), and heat of crystallization (∆H_c_). The crystallization temperature and the heat of crystallization are determined from the cooling cycle (exothermal peak). T_m_ is calculated based on the second heating cycle (Figure 7b). According to Table 3, the addition of Al narrows the crystallization window (∆T = T_onset_ − T_peak_) because Al does not hinder the LLDPE chain movements and slightly increases the crystallization progress.

The presence of Al increases the heat of crystallization (∆H_c_) (Table 3). As anticipated, the composite’s DSC endotherm displays one distinct major peak (T_m_) (Figure 7b). The endotherms of the PE/Al samples displayed a comparable characteristic [26]. Furthermore, the effect of Al content on the LLDPE/CaCO_3_ melting point is carefully observed. The effect of the filler on the thermal motion of the polymer can be explained by the fact that as the amount of Al in the LLDPE/CaCO_3_ composite increased, the T_m_ values slightly decreased and its intensity changed [27].

The thermal insulation performance of the Al/LLDPE/CaCO_3_ composite films is confirmed by the KES-F7 Thermo Labo II test method, and the results are presented in Table 4. The “coldness and warmth feeling” refers to the sensation of coldness or warmth when the skin makes contact with a fabric. Depending on how much heat is transferred from the skin to the fabric, a person may feel cold or warm. This feeling is measured using the KES-F7 Thermo Labo II test method by calculating the q_max_ value (peak heat flux). The results show that the film’s thermal insulation capacity increases after the addition of Al. The composite without Al possesses the lowest thermal insulation capacity (22%), whereas the composite with 8 wt.% Al shows the highest thermal insulation capacity (34.6%). Additionally, the thermal insulation of the composite film increases as the Al content increases. Good thermal insulation properties of Al were also reported by Zeng et al. [28], who explained that when an aluminum foil was added to the inner and outer surfaces of the insulating material, the maximum insulation time increased by 25.39% and 4.03%, respectively, compared with the insulating material without an aluminum foil. Putting the aluminum foil on the back of the double-sided insulating material significantly improves the thermal insulation performance.

Moisture permeability is the ability of a material to allow water vapor to pass through and is a key factor in the performance of a breathable film [29]. The water vapor transmission mechanism is known as water vapor diffusion by the interconnection between pores inside the film, and the expansion of the internal structure by stretching during processing improves moisture permeability. Table 5 lists the WVTR of the studied films, which obviously shows that the Al (8 wt.%)/LLDPE/CaCO_3_ sample has the highest value compared with the Al (2 wt.%)/LLDPE/CaCO_3_ and Al (4 wt.%)/LLDPE/CaCO_3_ samples because of the large size of the pores (average pore diameter = 4.13 µm) (Figure 3) within the polymer matrix. Additionally, the WVTR value of the composite film increases as the Al content increases.

Furthermore, because the water solubility in the crystal area is inherently lower than that in the amorphous area, the water solubility inside the polymer matrix decreased as the degree of crystallinity increased [30]. The effect of adding Al with LLDPE/CaCO_3_ on the WVTR is also carefully observed. A significant reduction in WVTR is observed for the composite films with 2 and 4 wt.% Al compared with the LLDPE/CaCO_3_ matrix. The free volume and diffusion path accessible to water diffusion decreases because of LLDPE/CaCO_3_ and Al immiscibility, which, in turn, decreases the WVTR.

The waterproofness of the porous composite films is determined by evaluating the water pressure for water penetration, as shown in Table 5. The porous membranes fabricated with 2 and 4 wt.% Al exhibit excellent waterproofness for water pressures of 1030 and 1076 mm H_2_O, respectively. Subsequently, with an increase in the Al content to 8 wt.%, the waterproofness is reduced to 807 mm H_2_O. This substantial drop in water pressure can be attributed to the significant increase in pore diameter (Figure 3). According to the Young–Laplace equation, water pressure and pore diameter are inversely proportional [31]. The primary determinants of the significant properties of the waterproof-breathable films are pore size and pore density.

Figure 8a displays the stress–strain curves of the Al/LLDPE/CaCO_3_ composite films in the machine direction (MD), and Table 6 presents their tensile properties. According to Figure 8a, the stress-at-break value of the LLDPE/CaCO_3_ composite film increases after incorporating Al (2–8 wt.%), and the composite with 8 wt.% of Al loading shows the highest stress-at-break (10.41 MPa). In this instance, the improvement in the tensile properties can be explained by considering the potential impact of Al on the molecular orientation during the extrusion along the MD, as well as the strong matrix–particle adhesion [32], therefore promoting the increase in the tensile strength of the composite film with the incorporation of the Al particles (2–8 wt.%).

On the other hand, the stress–strain curves demonstrated that the strain-at-break value of the LLDPE/CaCO_3_ film increases from 76.15% to 138.03% with the incorporation of Al (2–8 wt.%). The Al particles are well dispersed on LLDPE/CaCO_3_, which efficiently increases the fracture strain [33]. Improved mechanical performance of PE-based composites filled with aluminum powder was also reported by Mysiukiewicz and colleagues [26]. From Figure 8a, a strain-hardening mechanism is observed, which may be a consequence of the orientation of the crystalline structure of the polymer in the MD [34]. The composites containing varying Al concentrations (2–8 wt.%) showed various strain-hardening mechanisms compared to the LLDPE/CaCO_3_ film, which may be due to the influence of Al dispersion on the polymer chain orientation along the MD [35].

The cross direction (CD) is the direction parallel to the MD. Figure 8b displays the stress–strain curves of the Al/LLDPE/CaCO_3_ composite films with varying Al contents (0–8 wt.%) in the CD, and Table 6 summarizes their tensile characteristics. Figure 8b shows that, similar to the MD sample (Figure 8a), the stress-at-break value of the Al (2–8 wt.%)/LLDPE/CaCO_3_ composite films slightly increases and the Al (8 wt.%)/LLDPE/CaCO_3_ composite film exhibits the highest stress-at-break (2.67 MPa). In this case, the improvement in the tensile properties can be explained by considering the probable influence of Al on the molecular orientation during the film extrusion, along with the good matrix–particle adhesion [32].

In contrast, the stress–strain curves demonstrated that the strain-at-break value of the LLDPE/CaCO_3_ composite film decreased from 217.94% to 139.35% with the incorporation of Al (2–8 wt.%). The Al particles may be stuck inside the polymer chain entanglements; consequently, the polymer’s total chain mobility is constrained [36]. Additionally, the mechanical properties of the LLDPE/CaCO_3_ composite may be affected by the presence of Al in the dispersion [34], as well as the various porous conditions of the composite films [37]. However, the 2 to 8 wt.% Al particles show good tensile properties (Table 6). In this study, the Al particles were well dispersed, presenting excellent reinforcing efficiency even at higher Al content (8 wt.%). As shown in Figure 8, the MD film performs better than the CD film, which may be a result of particles spreading in a preferential position during extrusion.

## 4. Conclusions

Thermal-insulating breathable LLDPE/CaCO_3_ composite films containing highly thermal-insulating Al particles (2–8 wt.%) were successfully developed using the extrusion molding technique at a pilot scale, and the effect of different Al contents on the material properties of the composite films was investigated. The SEM images revealed numerous irregular cracks and microsized pores in the film structure. CaCO_3_ was confirmed to be evenly dispersed on the film surface with an average particle size of 3.00 µm. As the Al content increased, the micropore size of the film increased, and stretching during composite film manufacturing using the extrusion molding process caused cracks to occur. Thus, the film’s ultimate properties were significantly influenced by particle dispersion and strong extrusion effect. Additionally, the tensile properties were influenced by the addition of Al: With the increase in Al content, the tensile properties increased, but the flexibility of the composite films was reduced in the CD. Moreover, the maximum stress-at-break was obtained for the composite film with 8 wt.%. Al. In addition, the TGA graph indicates that the inclusion of Al increases thermal stability, as evidenced by the decomposition temperature graphs of the composite rising at higher temperatures. After the incorporation of Al, other significant properties of the produced composite films, such as thermal insulation, waterproofness, and water vapor transmission capabilities, also increased. Finally, it is proven that thermal-insulating breathable composite films with high mechanical, thermal, and crystalline properties can be obtained, which can be used for wooden house wrapping, electronics, textiles, and packaging.

## Figures and Tables

**Figure 1 materials-16-04230-f001:**
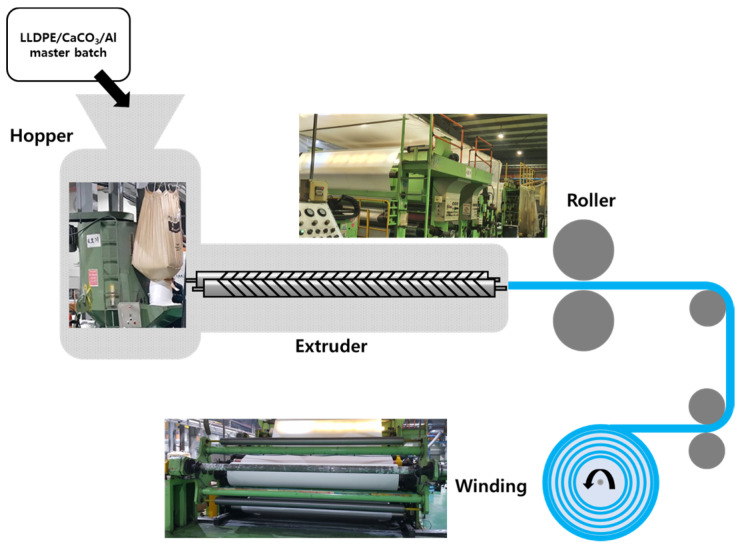
Illustration of the preparation of the Al/LLDPE/CaCO_3_ composite films.

**Figure 2 materials-16-04230-f002:**
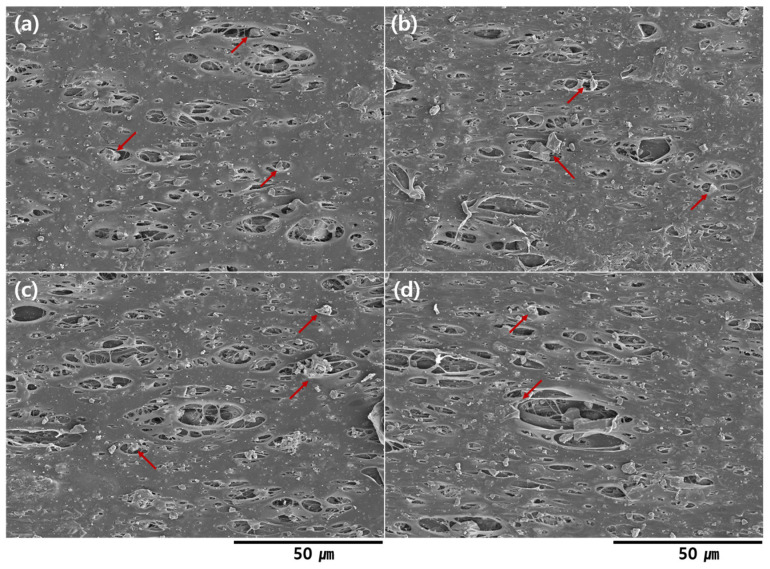
Field-emission scanning electron microscopy images of the Al/LLDPE/CaCO_3_ composite films with varying Al contents: (**a**) 0 wt.%; (**b**) 2 wt.%; (**c**) 4 wt.%; and (**d**) 8 wt.% (red arrow = CaCO_3_ particle).

**Figure 3 materials-16-04230-f003:**
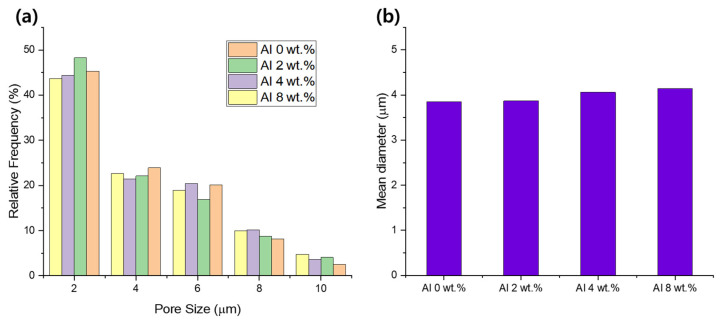
(**a**) Pore-size distribution histograms, and (**b**) mean pore diameter of the Al/LLDPE/CaCO_3_ composite films.

**Figure 4 materials-16-04230-f004:**
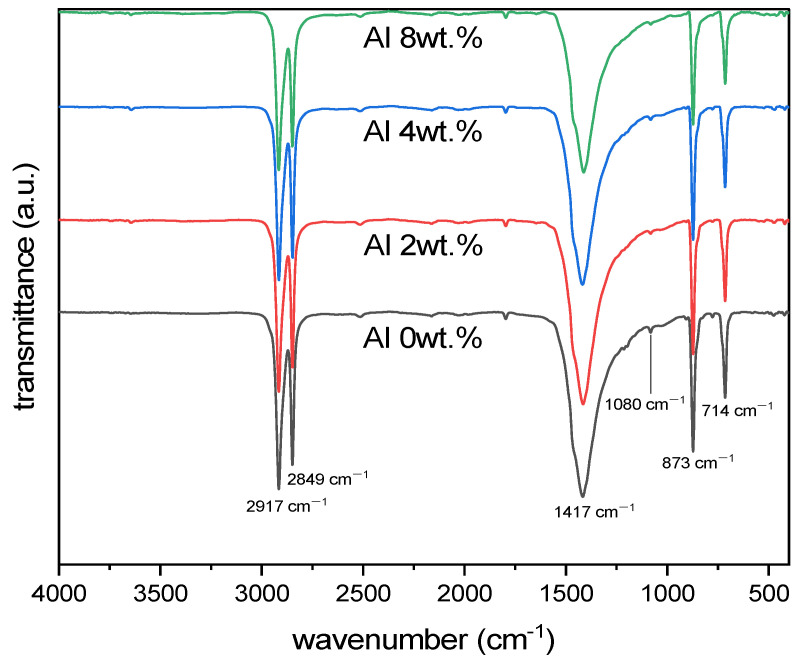
Data of the Al/LLDPE/CaCO_3_ composite films obtained by Fourier-transform infrared spectroscopy.

**Figure 5 materials-16-04230-f005:**
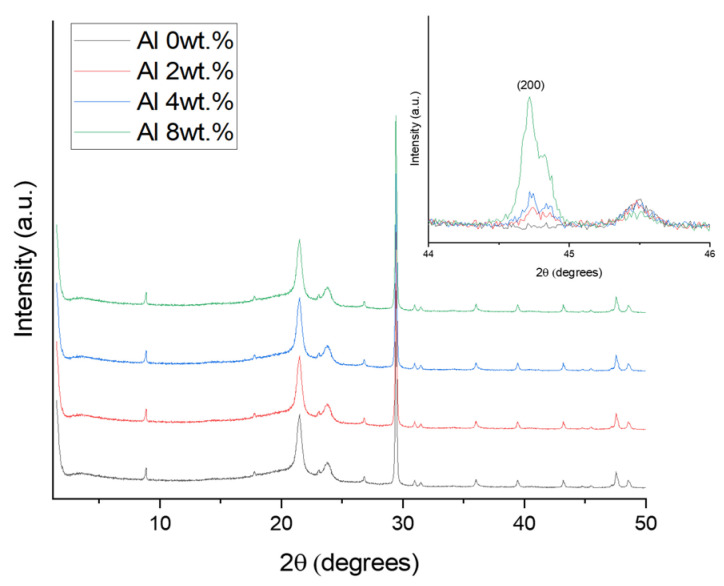
X-ray diffraction data of the Al/LLDPE/CaCO_3_ composite films.

**Figure 6 materials-16-04230-f006:**
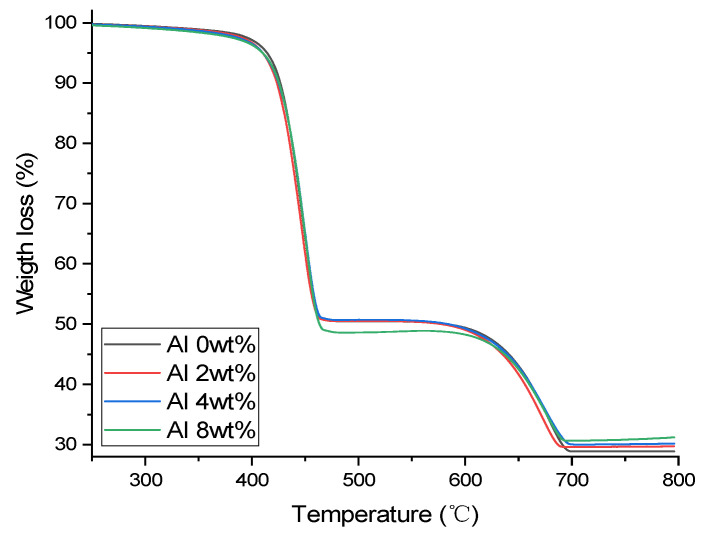
Thermogravimetric analysis results of the Al/LLDPE/CaCO_3_ composite films.

**Figure 7 materials-16-04230-f007:**
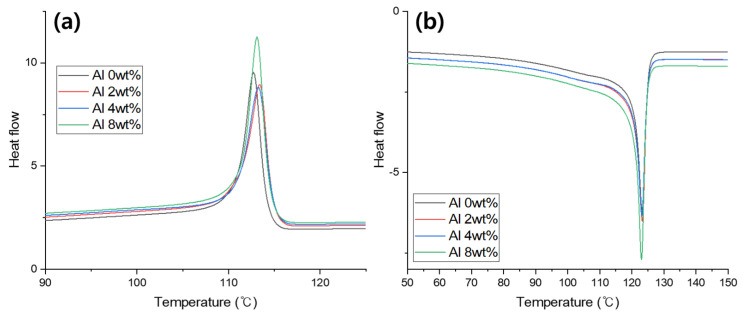
(**a**) Cooling curves and (**b**) heating curves of the Al/LLDPE/CaCO_3_ composite films obtained by differential scanning calorimetry.

**Figure 8 materials-16-04230-f008:**
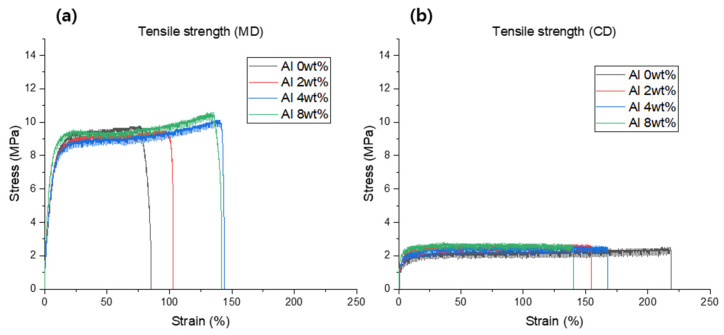
Stress–strain curves of the Al/LLDPE/CaCO_3_ composite films. (**a**) The samples were cut along MD, and (**b**) samples were cut along CD.

**Table 1 materials-16-04230-t001:** Material characteristics of linear low-density polyethylene (LLDPE).

Name	Grade	Melt Flow Index	Density	Melting Point	Haze (%)	Gloss
LLDPE	CEFOR 1221P	2.0 g/10 min	0.918 g/cm^3^	116 °C	0.56	151

**Table 2 materials-16-04230-t002:** Manufacturing process parameters.

Contents	Subsection
Extrusion temperature (°C)	120
Extrusion speed (rpm)	200
Stretching temperature (°C)	70
Stretching ratio	2.5

**Table 3 materials-16-04230-t003:** Material characteristics of the Al/LLDPE/CaCO_3_ composite films.

Sample Type	T_onset_ (°C)	T_endset_ (°C)	T_peak_ (°C)	∆T (°C)	T_m_ (°C)	∆H_c_ (J/g)
Al 0 wt.%	116.00	76.24	112.71	3.69	123.24	57.20
Al 2 wt.%	116.92	72.93	113.45	3.47	123.28	58.82
Al 4 wt.%	116.31	72.42	113.19	3.12	123.08	58.31
Al 8 wt.%	116.31	75.57	113.10	3.21	123.03	63.54

**Table 4 materials-16-04230-t004:** Thermal insulation results of the Al/LLDPE/CaCO_3_ composite films.

Sample Type	Thermal Insulation (%)
Al 0 wt.%	22.9 ± 0.2
Al 2 wt.%	34.2 ± 0.3
Al 4 wt.%	34.3 ± 0.3
Al 8 wt.%	34.6 ± 0.3

**Table 5 materials-16-04230-t005:** Water vapor transmission rate (WVTR) and water pressure of the Al/LLDPE/CaCO_3_ composite films.

Sample Type	WVTR (g/m^2^/24 h)	Water Pressure (mm H_2_O)
Al 0 wt.%	6653.93 ± 45	983.5 ± 2.5
Al 2 wt.%	6564.76 ± 42	1030 ± 3.0
Al 4 wt.%	6603.52 ± 39	1076 ± 20.5
Al 8 wt.%	6713.38 ± 43	807 ± 53.0

**Table 6 materials-16-04230-t006:** Stress–strain data of the Al/LLDPE/CaCO_3_ composite films.

Sample Type	Thickness (mm)	Stress-At-Break (MPa)		Strain-At-Break (%)	
		MD	CD	MD	CD
Al 0 wt.%	0.05	9.75	2.41	76.15	217.94
Al 2 wt.%	0.05	9.36	2.62	97.28	154.18
Al 4 wt.%	0.05	10.02	2.52	138.03	167.03
Al 8 wt.%	0.05	10.41	2.67	133.81	139.35

## Data Availability

Not applicable.

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
