# Peer review of "Preparation and Characterization of Thermal-Insulating Microporous Breathable Al/LLDPE/CaCO3 Composite Films"

_materials, 2023, doi:10.3390/ma16124230_

Round 1

Reviewer 1 Report

1.     The abstract is too long. Please revised it to be more concise.

2.     The English of this work requires improving: A lot of sentences are hard to follow, like: Lines 137-138, “Considering the representative SEM images (figure 2) of the LLDPE/CaCO3/Al (0-8wt.%) microvoid generation was observed.”; Lines 319-321 “According to Figure 7(b) like MD sample (figure 7(a)I) the stress at break value of the LLDPE/CaCO3/Al film slightly increased after adding the Al content (2–8 wt.%) and the composite with 8 wt.% of Al loading exhibited the highest stress at break (2.67 MPa).”, Lines

3.     “LLDPE/CaCO3/Al (0-8 wt.%) composites” is used in this work. For a certain case, it is hard to express in a brief manner. How about “(0-8 wt.%)Al/LLDPE/CaCO3 composites”? Sometimes, the “8 wt.% Al/LLDPE/CaCO3 composite” can be used. This is more common in the field of composites.

4.     Please labelling four images in Figure 2 with a, b, c, and d.

5.     Lines 138-139, “As the Al content increased, it was observed that the micropore size of the film increased”. This claim should be supported by quantitative values.

6.     “The average particle size of CaCO3 is 3.00 ㎛, and it can be confirmed that it is uniformly dispersed on the film surface.” Where are the particles? Labelling them please.

7.     Line 157 “2 wt.% Al sample”; Line 158 “among 4 and 8wt.%, 4wt.% shows”; Line 173 “Al composites”; Line 245 “LLDPE sample”; Please unify the name of materials in a concise and tidy manner. Randomly use of “sample, composite or other name” is not preferrable.

8.     In Figure 3, the authors should label the characteristic peaks to guide the readers. At least, where are the different peaks for the films with and without Al addition?

9.     In Figure 4, the phases are not labelled in these XRD spectra.

10.   In Table 3, what are the units for T related factors?

1.     The abstract is too long. Please revised it to be more concise.

2.     The English of this work requires improving: A lot of sentences are hard to follow, like: Lines 137-138, “Considering the representative SEM images (figure 2) of the LLDPE/CaCO3/Al (0-8wt.%) microvoid generation was observed.”; Lines 319-321 “According to Figure 7(b) like MD sample (figure 7(a)I) the stress at break value of the LLDPE/CaCO3/Al film slightly increased after adding the Al content (2–8 wt.%) and the composite with 8 wt.% of Al loading exhibited the highest stress at break (2.67 MPa).”, Lines

3.     “LLDPE/CaCO3/Al (0-8 wt.%) composites” is used in this work. For a certain case, it is hard to express in a brief manner. How about “(0-8 wt.%)Al/LLDPE/CaCO3 composites”? Sometimes, the “8 wt.% Al/LLDPE/CaCO3 composite” can be used. This is more common in the field of composites.

4.     Please labelling four images in Figure 2 with a, b, c, and d.

5.     Lines 138-139, “As the Al content increased, it was observed that the micropore size of the film increased”. This claim should be supported by quantitative values.

6.     “The average particle size of CaCO3 is 3.00 ㎛, and it can be confirmed that it is uniformly dispersed on the film surface.” Where are the particles? Labelling them please.

7.     Line 157 “2 wt.% Al sample”; Line 158 “among 4 and 8wt.%, 4wt.% shows”; Line 173 “Al composites”; Line 245 “LLDPE sample”; Please unify the name of materials in a concise and tidy manner. Randomly use of “sample, composite or other name” is not preferrable.

8.     In Figure 3, the authors should label the characteristic peaks to guide the readers. At least, where are the different peaks for the films with and without Al addition?

9.     In Figure 4, the phases are not labelled in these XRD spectra.

10.   In Table 3, what are the units for T related factors?

Author Response

Dear Reviewer 1

We value the time you and the reviewers spent reading our work and providing valuable comments. Your valuable and incisive remarks inspired potential changes to the present edition. The authors have given great thought to the comments and have done our best to respond to each one. We hope the manuscript after thorough edits meet your high standards. Any more helpful feedback will be appreciated by the authors. The detailed responses are provided below. Revisions to the manuscript are marked up using the “Track Changes” function

1.The abstract is too long. Please revised it to be more concise.

Answer: The concise version of abstract has been made.

2.The English of this work requires improving: A lot of sentences are hard to follow, like: Lines 137-138, “Considering the representative SEM images (figure 2) of the LLDPE/CaCO3/Al (0-8wt.%) microvoid generation was observed.”; Lines 319-321 “According to Figure 7(b) like MD sample (figure 7(a)I) the stress at break value of the LLDPE/CaCO3/Al film slightly increased after adding the Al content (2–8 wt.%) and the composite with 8 wt.% of Al loading exhibited the highest stress at break (2.67 MPa).”, Lines

Answer: The manuscript is thoroughly reviewed, and where necessary, the English is improved.

  1. “LLDPE/CaCO3/Al (0-8 wt.%) composites” is used in this work. For a certain case, it is hard to express in a brief manner. How about “(0-8 wt.%)Al/LLDPE/CaCO3 composites”? Sometimes, the “8 wt.% Al/LLDPE/CaCO3 composite” can be used. This is more common in the field of composites.

Answer: In place of LLDPE/CaCO3/Al (0-8 wt.%), (0-8 wt.%)Al/LLDPE/CaCO3 is written.

  1. Please labelling four images in Figure 2 with a, b, c, and d.

Answer: Four images in figure 2 were given the labels a, b, c, and d.

  1. Lines 138-139, “As the Al content increased, it was observed that the micropore size of the film increased”. This claim should be supported by quantitative values.

Answer: To support the previously stated claim, numerical data is added.

  1. “The average particle size of CaCO3 is 3.00 ãŽ›, and it can be confirmed that it is uniformly dispersed on the film surface.” Where are the particles? Labelling them please.

Answer: In figure 2, the CaCO3 particles are labeled.

  1. Line 157 “2 wt.% Al sample”; Line 158 “among 4 and 8wt.%, 4wt.% shows”; Line 173 “Al composites”; Line 245 “LLDPE sample”; Please unify the name of materials in a concise and tidy manner. Randomly use of “sample, composite or other name” is not preferrable.

Answer: The material names are unified.

  1. In Figure 3, the authors should label the characteristic peaks to guide the readers. At least, where are the different peaks for the films with and without Al addition?

Answer: There are no clear difference between peak of matrix and composite that’s why only matrix peaks are labeled in figure 3. (Figure 3 is now updated as figure 4).

  1. In Figure 4, the phases are not labelled in these XRD spectra.

Answer: In figure 4, plane of Al peak is labelled. (Figure 4 is now updated as figure 5).

  1. In Table 3, what are the units for T related factors?

Answer: The units of T related factors are added in Table 3.

Reviewer 2 Report

The manuscript is devoted to the preparation and characterization of breathable Films prepared by extrusion molding based on linear low-density polyethylene (LLDPE) / Calcium carbonate (CaCO3) and Aluminium (Al; 0, 2, 4, and 8 14 wt.%). These films transmit moist vapor (breathability) through pores while maintaining a barrier to liquids. The addition of Al improves the breathability, thermal stability, and mechanical properties of such film up to level to be used for wood house wrapping, electronics packaging, etc.

The manuscript is interesting from the scientific point of view and particularly due to possible wider application.

 Before my recommendation for publication, typos should be corrected and some sentences should be reformulated.

20 “…composites were investigated using differential scanning calorimetry, which revealed that the presence of Al increases the crystallinity…”

Differential scanning calorimetry is not a method for estimating crystallinity. I found the definition “DSC detects endothermic and exothermic transitions like the determination of transformation temperatures and enthalpy of solids and liquids as a function of temperature. It is used for determination of glas transitions and the investigation of chemical reactions, melting and crystallization behavior”.

The sentence should be reformulated.

66 “…composite polymeric Films, Planes et al. [10]… “

Typo.

187 “…However, the LLDPE/CaCO3/Al composite film's XRD pattern (Figure 4) showed no obvious peaks of Al, proving that the Al was completely exfoliated.”

What do authors want to say?

203 “…degradation behavior and degradation happened due to LLDPE and Al decomposition…”

Al decomposition? The sentence should be reformulated.

304 “…Therefore, promoting the rise in the tensile strength of the composite film with the incorporation of Al particles (0-8 wt.%).”

Incomplete sentence

Author Response

Dear Reviewer 2

We value the time you and the reviewers spent reading our work and providing valuable comments. Your valuable and incisive remarks inspired potential changes to the present edition. The authors have given great thought to the comments and have done our best to respond to each one. We hope the manuscript after thorough edits meet your high standards. Any more helpful feedback will be appreciated by the authors. The detailed responses are provided below. Revisions to the manuscript are marked up using the “Track Changes” function

Reviewer 2

The manuscript is devoted to the preparation and characterization of breathable Films prepared by extrusion molding based on linear low-density polyethylene (LLDPE) / Calcium carbonate (CaCO3) and Aluminium (Al; 0, 2, 4, and 8 14 wt.%). These films transmit moist vapor (breathability) through pores while maintaining a barrier to liquids. The addition of Al improves the breathability, thermal stability, and mechanical properties of such film up to level to be used for wood house wrapping, electronics packaging, etc.

The manuscript is interesting from the scientific point of view and particularly due to possible wider application.  Before my recommendation for publication, typos should be corrected and some sentences should be reformulated.

20 “…composites were investigated using differential scanning calorimetry, which revealed that the presence of Al increases the crystallinity…”

Differential scanning calorimetry is not a method for estimating crystallinity. I found the definition “DSC detects endothermic and exothermic transitions like the determination of transformation temperatures and enthalpy of solids and liquids as a function of temperature. It is used for determination of glas transitions and the investigation of chemical reactions, melting and crystallization behavior”.

The sentence should be reformulated.

Answer: The DSC description's crystallinity section has been updated.

 66 “…composite polymeric Films, Planes et al. [10]… “

Typo.

Answer: Revised

187 “…However, the LLDPE/CaCO3/Al composite film's XRD pattern (Figure 4) showed no obvious peaks of Al, proving that the Al was completely exfoliated.”

What do authors want to say?

Answer: This sentence is revised.

203 “…degradation behavior and degradation happened due to LLDPE and Al decomposition…”

Al decomposition? The sentence should be reformulated.

Answer: The sentence is revised.

 304 “…Therefore, promoting the rise in the tensile strength of the composite film with the incorporation of Al particles (0-8 wt.%).”

 Incomplete sentence

Answer: The sentence is revised.

Reviewer 3 Report

The paper "Preparation and Characterization of thermal-insulating, microporous, breathable LLDPE/CaCO3/Al composite film" by Jungeon Lee, Sabina Yeasmin, Jae Hoon Jung, Tae Young Kim, Tae Yeong Kwon, Da Yeong Kwon and Jeong Hyun Yeum, is devoted to the study of the effect of aluminum on the properties of the LLDPE/CaCO3/Al composite. The kinetics of crystallization of composites was studied, while the concentration of CaCO3 was always maintained constant, the effect of aluminum on the state and morphology of the matrix, permeability and thermal insulation, and mechanical properties of the films were determined. The work has a high level of scientific professionalism, primarily due to the use of modern research methods. However, a few remarks need to be made:

1. The purpose of the work is not defined in the introduction and the objectives of the study are not formulated.

2. When studying the state and morphology of composites, there is no confirmation that CaCO3 particles are uniformly distributed over the film surface.

3. Fig. 2 does not illustrate the statement about interconnected pores. What is the porosity of samples with different Al content? In addition, it is difficult to accept the statement from the figure that an increase in the aluminum content contributes to an increase in the size of micropores and cracks, since numerical changes in the sizes of pores and cracks depending on the Al content are not indicated, there are no histograms of the distribution of pores by size from the Al content. This is an important point in the article, since the properties of the studied films will be affected by their structural characteristics, pore sizes, their relationship, and pore size distribution. Moreover, in the future, the authors explain the changes in many properties of the films on the size factor of micropores.

4. According to the results of X-ray diffraction analysis, the addition of aluminum did not significantly affect the overall structure of the matrix. Although, on the other hand, the authors show that aluminum additions lead to a change in the structure - with an increase in the aluminum content, the number and size of pores in the matrix increase. How to explain it?

5. The dependence of the water vapor propagation velocity (WVTR) and water pressure on the type of samples with different aluminum content is not monotonous (Table 5). This dependence is explained by the authors by the size of pores and cracks, which is not proof without providing numerical structural data for the samples.

6. The errors of experimental measurements of all properties of the samples are not indicated. For example, from Fig. 5, the assertion that Al additions lead to an increase in thermal stability is not conclusive.

7. How can one explain the anomalous behavior of all properties of films containing 4% Al?

8. Table 6 shows the thermal insulation results of composite films as a function of aluminum content. The composite with 8% Al shows a thermal insulation capacity (34.6%) similar to the film with 2% Al (34.2%). Composite with 4% Al has 31.6% thermal insulation capacity. The explanation of the authors of such a dependence on the size of micropores is not clear. How can this anomalous dependence be explained?

The remarks made do not diminish the significance of the work. The article can be recommended for publication after its completion.

Author Response

Dear Reviewer 3

We value the time you and the reviewers spent reading our work and providing valuable comments. Your valuable and incisive remarks inspired potential changes to the present edition. The authors have given great thought to the comments and have done our best to respond to each one. We hope the manuscript after thorough edits meet your high standards. Any more helpful feedback will be appreciated by the authors. The detailed responses are provided below. Revisions to the manuscript are marked up using the “Track Changes” function

  1. The purpose of the work is not defined in the introduction and the objectives of the study are not formulated.

Answer: The purpose of the work is now defined in the introduction.

  1. When studying the state and morphology of composites, there is no confirmation that CaCO3 particles are uniformly distributed over the film surface.

Answer: In figure 2, the CaCO3 particles are labeled.

  1. Fig. 2 does not illustrate the statement about interconnected pores. What is the porosity of samples with different Al content? In addition, it is difficult to accept the statement from the figure that an increase in the aluminum content contributes to an increase in the size of micropores and cracks, since numerical changes in the sizes of pores and cracks depending on the Al content are not indicated, there are no histograms of the distribution of pores by size from the Al content. This is an important point in the article, since the properties of the studied films will be affected by their structural characteristics, pore sizes, their relationship, and pore size distribution. Moreover, in the future, the authors explain the changes in many properties of the films on the size factor of micropores

Answer: Histograms of the distribution of pores and numerical value of pores is added in the figure 3, and line related to interconnected pores is revised.

  1. According to the results of X-ray diffraction analysis, the addition of aluminum did not significantly affect the overall structure of the matrix. Although, on the other hand, the authors show that aluminum additions lead to a change in the structure - with an increase in the aluminum content, the number and size of pores in the matrix increase. How to explain it?

Answer: Mentioned line related to matrix structure is revised and Al peak is mentioned in XRD results.

  1. The dependence of the water vapor propagation velocity (WVTR) and water pressure on the type of samples with different aluminum content is not monotonous (Table 5). This dependence is explained by the authors by the size of pores and cracks, which is not proof without providing numerical structural data for the samples.

Answer: Numerical data of pores are provided.

  1. The errors of experimental measurements of all properties of the samples are not indicated. For example, from Fig. 5, the assertion that Al additions lead to an increase in thermal stability is not conclusive.

Answer: Temperature range is added to show the thermal stability specifically. For other experiments error value is mentioned.

  1. How can one explain the anomalous behavior of all properties of films containing 4% Al?

Answer: Analysis (WVTR and thermal insulation) is again performed and results is revised. Manuscript is also revised accordingly.

  1. Table 6 shows the thermal insulation results of composite films as a function of aluminum content. The composite with 8% Al shows a thermal insulation capacity (34.6%) similar to the film with 2% Al (34.2%). Composite with 4% Al has 31.6% thermal insulation capacity. The explanation of the authors of such a dependence on the size of micropores is not clear. How can this anomalous dependence be explained?

Answer: Thermal insulation analysis is again performed and results is revised. Manuscript is also revised accordingly.

Round 2

Reviewer 1 Report

Accept in present form